# Influence of Laminin Coating on the Autologous In Vivo Recellularization of Decellularized Vascular Protheses

**DOI:** 10.3390/ma12203351

**Published:** 2019-10-15

**Authors:** Mahfuza Toshmatova, Sentaro Nakanishi, Yukiharu Sugimura, Vera Schmidt, Artur Lichtenberg, Alexander Assmann, Payam Akhyari

**Affiliations:** Department of Cardiovascular Surgery and Research Group for Experimental Surgery, Medical Faculty, Heinrich Heine University, 40225 Düsseldorf, Germany; Mahfuza.Toshmatova@med.uni-duesseldorf.de (M.T.); snakanishi@asahikawa-med.ac.jp (S.N.); Yukiharu.Sugimura@med.uni-duesseldorf.de (Y.S.); Vera.Schmidt@med.uni-duesseldorf.de (V.S.); Artur.LIchtenberg@med.uni-duesseldorf.de (A.L.); payam.akhyari@med.uni-duesseldorf.de (P.A.)

**Keywords:** bioengineering, laminin, coating, decellularization

## Abstract

Decellularization of non-autologous biological implants reduces the immune response against foreign tissue. Striving for in vivo repopulation of aortic prostheses with autologous cells, thereby improving the graft biocompatibility, we examined surface coating with laminin in a standardized rat implantation model. Detergent-decellularized aortic grafts from donor rats (n = 37) were coated with laminin and systemically implanted into Wistar rats. Uncoated implants served as controls. Implant re-colonization and remodeling were examined by scanning electron microscopy (n = 10), histology and immunohistology (n = 18). Laminin coating persisted over eight weeks. Two weeks after implantation, no relevant neoendothelium formation was observed, whereas it was covering the whole grafts after eight weeks, with a significant acceleration in the laminin group (p = 0.0048). Remarkably, the intima-to-media ratio, indicating adverse hyperplasia, was significantly diminished in the laminin group (p = 0.0149). No intergroup difference was detected in terms of medial recellularization (p = 0.2577). Alpha-smooth muscle actin-positive cells originating from the adventitial surface invaded the media in both groups to a similar extent. The amount of calcifying hydroxyapatite deposition in the intima and the media did not differ between the groups. Inflammatory cell markers (CD3 and CD68) proved negative in coated as well as uncoated decellularized implants. The coating of decellularized aortic implants with bioactive laminin caused an acceleration of the autologous recellularization and a reduction of the intima hyperplasia. Thereby, laminin coating seems to be a promising strategy to enhance the biocompatibility of tissue-engineered vascular implants.

## 1. Introduction

Cardiovascular disease is the main cause of death globally [1]. In case of need for small-caliber arterial replacement, artificial implants have proven to be insufficient, and autologous grafts are frequently limited in terms of wall quality, availability, and long-term durability.

Particularly in the last decade, tissue-engineered large-caliber arterial and valvular grafts have presented good patency and mid-term durability in preclinical animal models as well as in humans [2,3,4]. For small-caliber arterial grafts, decellularization has been shown to improve their performance in small animal models as well. Decellularization reduces the inflammatory response against non-autologous implants [5]. Depending on the agents that are used to obtain acellular scaffolds, the speed of cellular repopulation varies [6]. Implant coating with bioactive proteins can further accelerate the repopulation process [7,8,9]. Not only beneficial results, such as rapid re-endothelialization and cell migration into the media without any inflammatory reactions, were observed in coated grafts, but also adverse intima hyperplasia occurred [7,8].

Laminins are heterotrimeric glycoproteins of the extracellular matrix that especially occur in the basement membrane [10]. They can bind to other matrix molecules, thereby contributing to cell differentiation, cell shaping and migration, maintenance of tissue phenotypes and promotion of tissue survival [11]. Laminins are crucial components for basement membrane assembly, initiating the process by binding to surface receptors and receptor-like molecules. Furthermore, laminins participate in the assembly of the cytoskeleton, promoting cell migration, adhesion of epithelial cells and hemidesmosome formation by their cytoplasmic domains [12].

In our study, we aimed to accelerate the non-hyperplastic autologous in vivo recellularization of decellularized aortic grafts using laminin for biofunctional implant coating.

## 2. Materials and Methods 

### 2.1. Animals

Male Wistar rats (200–250 g) from the animal care facility of the Heinrich Heine University (Duesseldorf, Germany) were used for all groups. All experiments were approved by the state animal care committee (reference number 84–02.04. 2012. A391) and conducted following the national animal welfare act.

### 2.2. Preparation of Donor Aorta and Graft Decellularization

Aortic graft harvesting (n = 37) was conducted as recently published [6]. In brief, donor rats were euthanized by isoflurane. Following thoracotomy, the aorta was dissected from surrounding tissue, before thorough antegrade and retrograde rinsing (phosphate buffered saline (PBS)) with 12.5 IU/mL heparin was carried out. Afterwards, a U-shaped aortic graft was prepared.

Harvested grafts were decellularized according to a recently published process using a protocol employing only biologically derived components and consisting of: 3 days of cycles with 50 nM latrunculin in glucose D-MEM, 0.6 M KCl, 1.0 M KI, 1 kU/mL DNase I in PBS and 3 cycles of 24 hours with 1% penicillin/streptomycin and 0.5% sodium azide. The protocol was performed in 15 mL tubes, filled with 6 mL per graft, containing up to 2 grafts. Appendix A displays a representative graft decellularized by this protocol.

### 2.3. Graft Coating with Fluorescent Laminin

Grafts (n = 23) were incubated in 1 mg/ml laminin (Sigma Aldrich, Taufkirchen, Germany) in PBS for 24 hours at 37 °C. After incubation, the grafts were shortly rinsed with PBS and transferred to implantation. In order to examine the persistency of laminin coating, implantations of grafts (n = 9) with laminin coupled with Alexa Fluor 488 (Invitrogen, Carlsbad, CA, USA) were performed. For the coupling procedure, laminin solution (1 mg/mL in PBS) was added to a sodium bicarbonate buffer (1 M NaHCO3, pH 7.3) to a final concentration of 20 µM. Labeling was conducted in the presence of 20–fold molar excess of the fluorophore Alexa488 by incubation with rotation for 1 h at room temperature in the dark.

### 2.4. Graft Implantation

The implantation procedure was conducted as previously published [8]. Recipient rats were anesthetized by 2.0%–2.5% isoflurane inhalation, orally intubated, and central venous catheter insertion followed. After median laparotomy, the intestines were lateralized, and the abdominal aorta was dissected from the inferior vena cava. Heparin was administered systemically (300 IU/kg), the aorta was clamped, and a U-shaped, decellularized aortic graft was anastomosed to the infrarenal aorta using an end-to-side technique and 10–0 monofilament, nonabsorbable polypropylene sutures (Ethicon, Norderstedt, Germany). Intermittent reperfusion guaranteed a minimization of the limb ischemia times. After release of graft blood flow, the abdominal aorta between the anastomoses was ligated for improved perfusion (Appendix A). After clinical observation during reperfusion for at least 10 min, particularly paying attention to the lower extremities perfusion, the abdomen was closed, and recipients recovered from anesthesia. Immediately after the implantation, Doppler sonography assessment of the implants was conducted to evaluate their function (Philips HDX11 ultrasonography system equipped with a 15 MHz probe, Philips, Amsterdam, Netherlands).

### 2.5. Graft Explantation

Grafts were explanted after 2 and 8 weeks from recipient rats anesthetized as described above. Laparotomy was performed, the abdominal aorta was cannulated, and the implanted grafts were rinsed with heparinized PBS, thoroughly excised, and further processed for scanning electron microscopy (SEM) (n = 10) or histology (n = 27). Aortic grafts were divided into four regions: proximal anastomosis (region A1), ascending aorta (region A2), descending aorta (region B1) and distal anastomosis (region B2) (Appendix A). After 2 weeks in vivo, grafts were fixed in 2.5% glutaraldehyde solution for 1 h, washed in sodium chloride 0.9% solution 3 times for 5 min and dehydrated with ethanol (50%, 70%, 80%, 96%, 100%) for 5 min in each. After drying completely, the regions were gold-coated and then examined under a scanning electron microscope (Leo 1430 VP, Zeiss, Wetzlar, Deutschland).

### 2.6. Histology

Graft cryo-sections (5 µm) underwent histological assessment (hematoxylin/eosin staining, von Kossa staining, Movat’s pentachrome staining). The amount of luminal neointima formation was examined by a standardized hematoxylin/eosin staining-based protocol as recently published. In brief, each graft was divided into four regions as described above, and three sections from each region were analyzed in eight segments divided by radial lines. In each segment, the percentage of neointima formation was determined. Similarly, in each of the eight graft wall segments, the media repopulation was evaluated as follows: The number of cells migrating to the implant media was counted. To measure the amount of adverse intima hyperplasia, the intima-to-media ratio was calculated in the same predefined areas and segments of the grafts as described above. In each segment, the thickness of the neointima and the media was measured to assess the intima-to-media ratio.

In order to determine the amount of calcification after explantation, cross-sections were stained using a standard protocol for von Kossa staining, and then subdivided into four pieces. The von Kossa score was calculated based on a scoring system representing the relative extent of von Kossa-positive areas, with intima values ranging from 0 to 3, while media calcification values ranged from 0 to 5 [13].

### 2.7. Immunohistology

Cryo-sections (5 µm) were incubated for 10 min with 0.25% triton-X and 1 h with 5% bovine serum albumin + 0.1% tween-20, in each case at room temperature. The primary antibodies were: anti-von Willebrand factor ((vWF), DAKO Hamburg, Germany), anti-alpha-smooth muscle actin ((aSMA), Sigma Aldrich, Taufkirchen, Germany), anti-CD3 (Sigma Aldrich, Taufkirchen, Germany), and anti-CD68 (Abcam, Cambridge, UK), each + 1% bovine serum albumin + 0.1% tween-20 for 1 h at 37 °C. Secondary antibodies conjugated to the fluorophores Alexa 546 and Alexa 488 (Invitrogen, Carlsbad, CA, USA) + 1% bovine serum albumin + 0.1% tween–20 were applied for 45 min under dark and humid conditions at 37 °C. Vectashield mounting medium containing DAPI (4’,6-Diamidino-2-phenylindol) was used to cover the sections, and microscopy was conducted utilizing a DM2000 system with a digital camera DFC 425C (Leica, Wetzlar, Germany) and the Leica Application Suite V3.7 software.

### 2.8. In Situ Zymography

To evaluate the MMP (matrix metalloproteinase) activity in explanted grafts, in situ zymography was performed. Cryo-sections (5 µm) of explanted aortic grafts were incubated for 20 h at room temperature with fluorescein-labeled gelatin (40 g/ml; Invitrogen, Carlsbad, CA, USA) in 50 mM Tris-HCl, 10 mM CaCl_2_, 150 mM NaCl and 5% triton-X. In order to examine the specificity of gelatinase activity, incubations with gelatin and 20 mM EDTA, only Buffer without gelatin and cis-ACCP solution (Cayman Chemicals, Ann Arbor, MI, USA) were conducted. Finally, sections were mounted with DAPI-containing Vectashield (Vector Labs, Peterborough, United Kingdom), and the MMP activity was visualized by fluorescence microscopy as described above. The MMP activity was assessed using Image J to measure the mean fluorescence intensity of the aortic graft wall.

### 2.9. Statistics

All continuous variables are presented as mean values ± standard errors of the mean. Student’s t-tests were conducted for direct group comparisons. p-values lower than 0.05 were considered to indicate statistical significance. Data analysis was performed with GraphPad Prism v 6.01 (GraphPad Software, San Diego, CA, USA).

## 3. Results

### 3.1. Operative Outcome

The graft harvesting time amounted to 2.3 ± 3.5 min, while the graft implantation cut-suture time was 46.2 ± 5.5 min with infrarenal aortic clamping times ranging from 12 to 31 min (17.3 ± 4.6 min).

### 3.2. Laminin Coating Persistency

Alexa Fluor 488-coupled laminin coating resulted in a continuous, brightly green fluorescence along both surfaces of the graft. The fluorescence was found to be persistent after two and eight weeks, though the intensity was visually reduced after eight weeks (Figure 1).

### 3.3. Quantity and Quality of Neointima Formation

After two weeks, only extracellular matrix was detected on the luminal surface of SEM samples from explanted decellularized grafts, while autologous cellular colonization was not observed (Figure 2).

Hematoxylin/eosin staining at 8 weeks showed neointima in the laminin group consisting of predominantly one layer or a completely restructured graft wall including fiber remodeling, mostly in A1 and B2 regions, whereas parts with multi-layer neointima were also observed. In the control group, neointima predominantly consisted of areas with multi-layer hyperplastic intima (Figure 3).

After 8 weeks in vivo, a continuous de novo cellular repopulation on the luminal side of decellularized conduits was observed, which was significantly higher in the laminin group (Percentage of re-endothelialized luminal surface: 98.4% ± 0.6% vs. 91.3% ± 3.1% in the control group, p = 0.0048).

For analysis of adverse hyperplastic neointima formation, the intima-to-media ratio was measured in explanted decellularized aortic grafts. Eight weeks after implantation, the overall intima-to-media ratio in the laminin group was significantly lower than in the uncoated controls (0.9 ± 0.1 vs. 1.5 ± 0.2, p = 0.0149) (Figure 4). In all subregions of the graft except in B1, the intima-to-media ratio in the laminin group was decreased as compared to controls.

Immunofluorescence staining of decellularized aortic grafts 8 weeks after implantation revealed most of the cells in the hyperplastic intima areas of both groups to contain aSMA. The luminal neoendothelial layer stained positive for vWF (Figure 5A,B). By Movat’s pentachrome staining, in hyperplastic intima areas, glycosaminoglycan-rich extracellular substance was detected around cells with a fibroblastoid phenotype. In both groups, no inflammatory cell markers (CD3 for lymphocytes and CD68 for macrophages) were detected at 8 weeks (Figure 5C,D).

After 8 weeks in vivo, von Kossa staining revealed small areas of microcalcification in the neointima and in the tunica media, and macrocalcification in the tunica media of the decellularized implants (Appendix A). Remodeled regions of the tunica media with high content of aSMa-positive autologous cells did not exhibit hydroxyapatite deposition. The extent of calcification, as assessed by the previously established von Kossa score, was not significantly different between the laminin and control groups, neither in the neointima (0.3 ± 0.1 in the laminin group vs. 0.5 ± 0.01 in the control group, p = 0.0661), nor in the media (1.7 ± 0.2 vs. 1.1 ± 0.2, p = 0.0779) (Figure 6).

### 3.4. Medial Graft Repopulation and Restructuring

The number of autologous cells repopulating the decellularized media was counted in all graft regions, and did not differ between the groups (Laminin: 127.6 ± 69.13 vs. Control: 242.2 ± 71.94 cells per cross-section, p = 0.2577). In the laminin group, remodeling of the media by autologous cells migration and extracellular matrix production was mostly observed in the A1 and B2 regions around the graft anastomoses, while in the control group, it was found more frequently only in the A1 region. The restructured media was full of fibroblast-shaped cells.

In situ zymography after 8 weeks showed MMP activity in the adventitia and predominantly in the neointima of decellularized aortic grafts. Areas of increased cell density, such as in hyperplastic neointima, exhibited a marked MMP activity, but there was no statistically significant difference between the two groups (p = 0.4170) (Figure 7).

## 4. Discussion

In the present study, we report on the impact of laminin coating on the in vivo fate of decellularized aortic grafts. In particular, laminin coating accelerated the autologous cellular repopulation of the implants, while inhibiting adverse intima hyperplasia.

For interpretation of the study results, the durability of laminin coating is important. In earlier reports, there was no information on the persistency of laminin coating in vivo, whereas in our study, laminin-bound fluorescence was found to persist on decellularized grafts for at least eight weeks. The green-fluorescent Alexa 488 coupled to the laminin proteins was initially observed along the surfaces of the grafts. However, after implantation in the blood circulation of rats, laminin spread through the whole graft wall, which may be influenced by blood pressure. Besides laminin movement through the tissue, separation and movement of the fluorophores may be considered as the only explanation for the observed distribution of the fluorescence signal over time. In this scenario, laminin might have been degraded earlier, while the fluorophore might persist in the graft wall. On the other side, laminin might persist to an even larger extent than was assumed from the detected fluorescence signal, since the fluorophore itself might have been degraded in vivo by enzymatic activity. Taken together, the kinetics of the in vivo degradation of fluorophores themselves as well as their uncoupling from bioactive proteins need to be further elucidated. Laminin antibody staining is not supportive in this context. We had previously conducted laminin antibody staining of decellularized and coated grafts, but all the grafts had stained positive, confirming data from other groups [6]. A reason for this issue may be the immunological relation of most laminin isoforms to laminin 111, since they contain either the β1 or the δ1 chains, or both. Therefore, antisera raised against laminin 111 purified from the EHS sarcoma stain all basement membranes, even in the absence of the laminin α1 chain. However, even in the case of potentially earlier degradation of laminin coating, its beneficial effects on graft repopulation have been observed during the follow-up until week 8.

In our study, laminin significantly accelerated the de novo endothelialization of decellularized grafts. The potential of laminin to induce the adhesion of circulating progenitor cells has been discussed previously. In vitro studies have shown that the human laminin α2 large globular 1 domain exhibits cell adhesion activity and binds to syndecan-1, which was proven in the cultured PC12 cell line from transplantable rat pheochromocytoma [14]. Laminin and laminin-derived peptides promote cell adhesion also in dental implants in vitro [15], and laminin-derived peptides contribute to the in vivo regeneration of peripheral nerves in rats. The re-endothelialization of acellular implants coated with laminin begins within the first two weeks after implantation, whereas relevant amounts of cellular population occur between week 2 and week 8. These data are in line with findings from previous cardiovascular graft implantation studies in the herein applied rat model [7].

In the present examination, laminin did not only accelerate re-endothelialization, but also decreased adverse neointima hyperplasia in decellularized aortic grafts. The anastomoses regions underwent an early remodeling process with high autologous cell migration activity, supporting the hypothesis that implant cellularization starts predominantly from the anastomoses parts as previously described [7,15,16]. The neointima areas were populated by aSMA-positive myofibroblasts, and none of the restructured parts of the grafts stained positive for inflammatory markers, implicating the invading cells most likely to be activated fibroblasts originating from the native aorta, and excluding relevant inflammatory response against the implants.

A previous in vitro study has reported on the migration-promoting activity of laminin isoforms in tumor cells [17]. Nevertheless, laminin surface coating of decellularized grafts did not significantly increase medial repopulation in the whole grafts after eight weeks in vivo, though aSMA-positive cells invasion was observed in the areas of media fiber restructuring. Since the repopulation of the media occurred mostly in the anastomotic parts and barely in remote parts of the grafts, longer follow-up periods may be necessary to observe complete media repopulation.

In spite of beneficial effects of laminin on neointima formation, including a trend towards reduced intima calcification, we could observe a tendency of laminin-coated grafts to develop media calcification. In this context, in vitro studies have indicated that laminin-1 can selectively recruit osteoprogenitors through an integrin β1-dependent cell attachment effect [18]. Moreover, previous in vivo examinations reported that laminin stimulates osteointegration [19]. The areas of wall restructuring did not show any calcification, whereas parts far from the anastomoses were more susceptible to calcification, which may be due to enhanced calcium binding to the extracellular matrix, particularly to damaged elastin fibers [20], in the absence of matrix-producing cells. Therefore, further strategies are necessary to promote an acceleration of autologous media repopulation, which may be addressed by cell-attracting agents as well as modified decellularization protocols resulting in a graft matrix architecture that favors cell migration [6].

## 5. Conclusions

Laminin coating significantly accelerated the non-inflammatory autologous in vivo recellularization and decreased the adverse neointima hyperplasia of decellularized aortic grafts. In this regard, laminin coating seems to be a promising strategy to improve the biocompatibility of tissue-engineered vascular implants, whereas a combination with alternative bioactive coating agents might further improve the outcome, particularly in terms of long-term calcification of the grafts.

## Figures and Tables

**Figure 1 materials-12-03351-f001:**
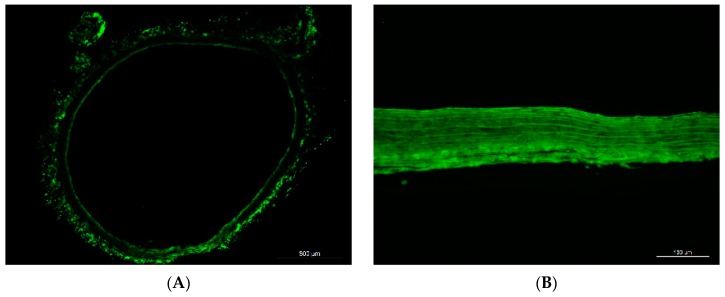
Laminin coating resulted in bright green fluorescence (Laminin-bound Alexa 488) along both surfaces of the aortic grafts (**A**). Two weeks after implantation, green fluorescence was detected throughout the whole graft wall (**B**). By week 8, the intensity of the fluorescence had decreased (**C**) Control graft 8 weeks after implantation (**D**). Blue in D, DAPI (not stained in (**A**–**C**)). Scale bars = 500 µm in (**A**); 100 µm in (**B**–**D**).

**Figure 2 materials-12-03351-f002:**
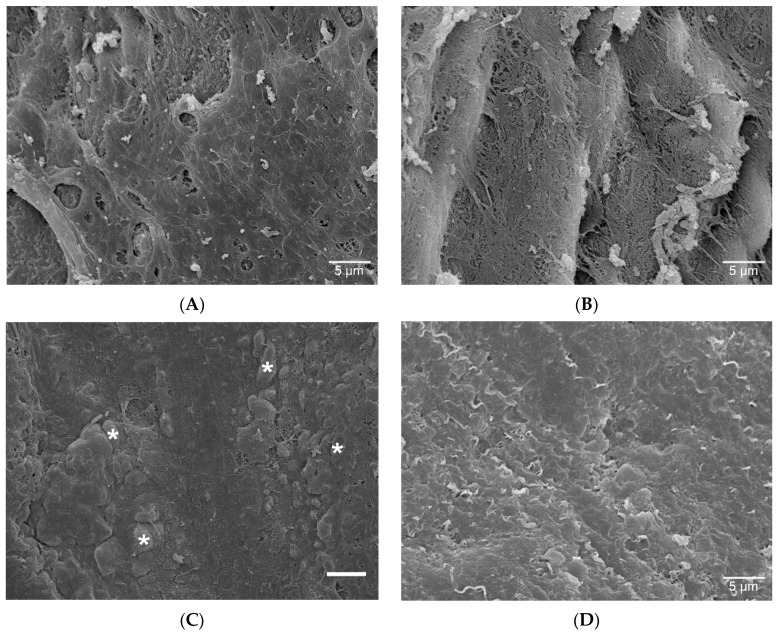
Representative SEM pictures of grafts after 2 weeks in vivo. Laminin group (**A**), Control group (**B**). Non-implanted controls: Native aorta (**C**) (exemplary asterisks on endothelial cells), and decellularized aorta (**D**). Scale bars = 5 µm.

**Figure 3 materials-12-03351-f003:**
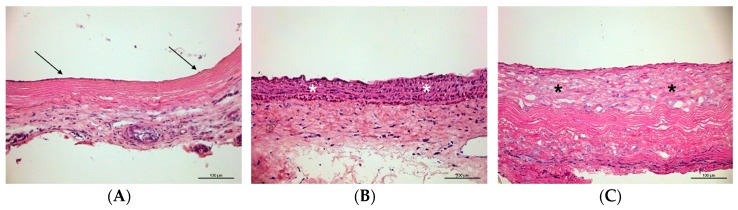
Cross-sections of grafts after 8 weeks in vivo. In the laminin group, the neointima predominantly consisted of one layer (arrows in (**A**)), and areas with restructured implant wall medially populated by autologous cells were observed (asterisks in (**B**)), whereas the neointima of the control group mostly exhibited intima hyperplasia (asterisks in (**C**)). Hematoxylin/eosin staining. Scale bars = 100 µm.

**Figure 4 materials-12-03351-f004:**
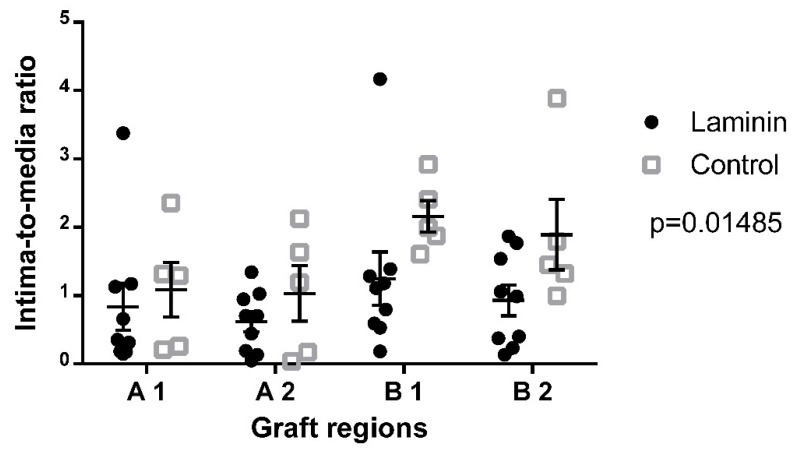
Semiquantitative analysis of the intima-to-media ratio 8 weeks after implantation showed significantly higher scores in the control group as compared to the laminin-coated group.

**Figure 5 materials-12-03351-f005:**
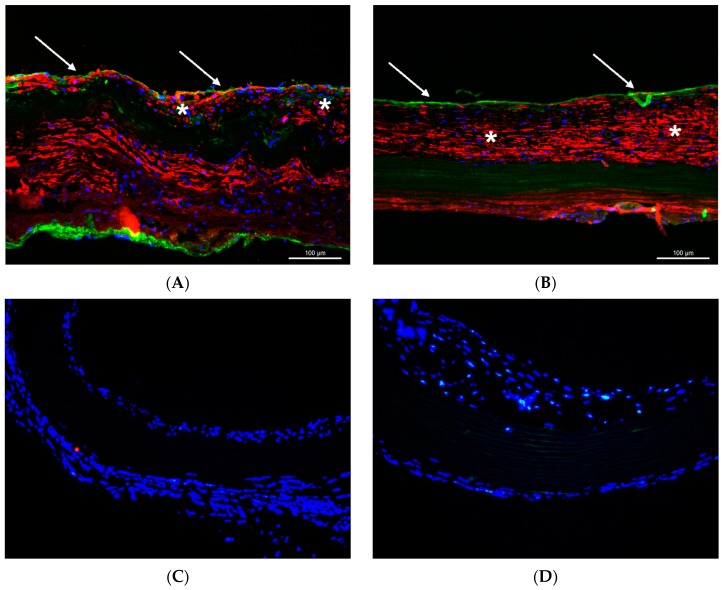
Representative sections through the ascending aorta of grafts after 8 weeks in vivo. In both groups (Laminin group: (**A**), Control group: (**B**)), the neointima (asterisks) was formed by aSMA-positive cells (red), and vWF-positive endothelial cells covered the luminal border (green, arrows). In both groups (Laminin group: (**C**), Control group: (**D**)), inflammatory markers did not stain positive for CD3 (red) or CD68 (green). Blue, DAPI. Scale bars = 100 µm in (**A**,**B**); 200 µm in (**C**,**D**).

**Figure 6 materials-12-03351-f006:**
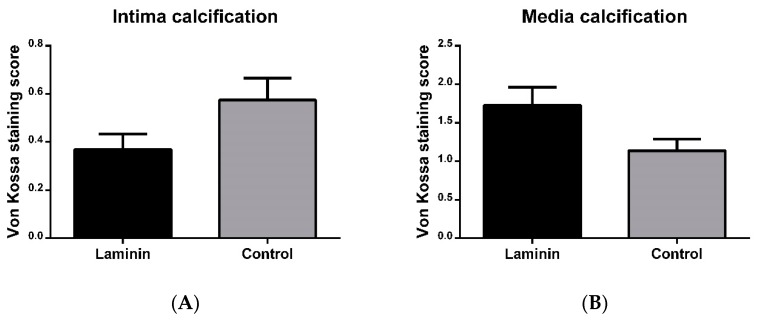
Semiquantitative analysis of intima (**A**) and media (**B**) calcification after 8 weeks in vivo. Neither in the intima nor in the media was the amount of calcification significantly different in laminin-coated versus uncoated control grafts.

**Figure 7 materials-12-03351-f007:**
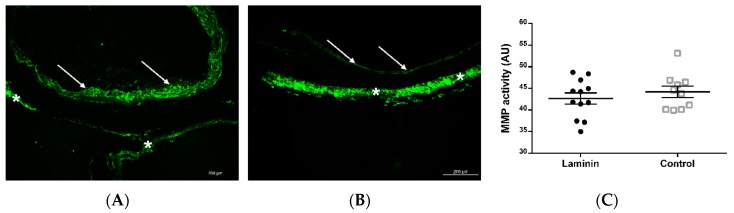
Representative cross-sections with in situ zymography after 8 weeks showed MMP activity (green) predominantly in the neointima (asterisks) and adventitia (arrows) in the control (**A**) and laminin (**B**) group, while semiquantitative analysis (**C**) did not show a statistically significant difference. AU = arbitrary units.

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
