# Peer review of "Influence of Laminin Coating on the Autologous In Vivo Recellularization of Decellularized Vascular Protheses"

_materials, 2019, doi:10.3390/ma12203351_

Round 1

Reviewer 1 Report

In this manuscript, the authors implemented laminin coating on decellularized aortic grafts and implanted the grafts to the infrarenal aorta in the rat model. The effects of laminin coating on recellularization, calcification and inflammation were studied. The method is simple and seems effective.

Major comments:

The authors claimed that the coated laminin was persistence after 8 week implantation. The conclusion was based on the fluorescent imaging. It is possible that the laminin was already degraded, leaving the fluorescent dye there. OR the fluorescence was even pseudo. It is hard to explain why the coated laminin with low concentration can be persistence for 8 weeks, even after host cells penetrating into the decellularized constructs. More discussion/explanation should be provided. For unknown reasons, the supplemental figures are not available. Some of the supplemental figures should be presented in the main text, like the inflammation data, MMP data, since they are actually important.

Author Response

Response to Reviewer 1

Reviewer’s Comment 1

In this manuscript, the authors implemented laminin coating on decellularized aortic grafts and implanted the grafts to the infrarenal aorta in the rat model. The effects of laminin coating on recellularization, calcification and inflammation were studied. The method is simple and seems effective.

Our Response:

We thank the reviewer for appreciation of our results. Following his general suggestion, the methods description has been structurally improved.

Reviewer’s Comment 2

The authors claimed that the coated laminin was persistence after 8 week implantation. The conclusion was based on the fluorescent imaging. It is possible that the laminin was already degraded, leaving the fluorescent dye there. OR the fluorescence was even pseudo. It is hard to explain why the coated laminin with low concentration can be persistence for 8 weeks, even after host cells penetrating into the decellularized constructs. More discussion/explanation should be provided.

Our Response:

We agree with the reviewer that the detection of persistence of laminin coating has to be critically discussed, what had been started in the discussion of our initial submission. However, since this is an important aspect, we have extended the discussion on this topic to further clarify potential explanations for our findings. The hypothesis of green pseudofluorescence can be ruled out, since negative controls without fluorescent laminin coating did not show green fluorescence under equal microscope settings.

Please find the following revised section in the second paragraph of the discussion: “For interpretation of the study results, the durability of laminin coating is important. In earlier reports, there was no information on the persistency of laminin coating in vivo, whereas in our study, laminin-bound fluorescence was found to persist on decellularized grafts for at least 8 weeks. The green-fluorescent Alexa 488 coupled to the laminin proteins was initially observed along the surfaces of the grafts, however, after implantation in the blood circulation of rats, laminin has spread through the whole graft wall, which might be influenced by blood pressure. Besides laminin movement through the tissue, separation and movement of the fluorophores only may be considered as an explanation for the observed distribution of the fluorescence signal over time. In this scenario, laminin might have been degraded earlier, while the fluorophore might persist in the graft wall. On the other side, laminin might persist to an even larger extent than it was assumed from the detected fluorescence signal, since the fluorophore itself might have been degraded in vivo by enzymatic activity. Taken together, the kinetics of the in vivo degradation of fluorophores themselves as well as their uncoupling from bioactive proteins need to be further elucidated. Laminin antibody staining is not supportive in this context. We had previously conducted laminin antibody staining of decellularized and coated grafts, but all the grafts had stained positive, confirming data from other groups [6]. A reason for this issue is that most laminin isoforms are immunologically related to laminin 111, since they contain either the β1 or the δ1 chains, or both. It explains why antisera raised against laminin 111 purified from the EHS sarcoma stain all basement membranes, even if the laminin α1 chain is absent. However, even in case of potentially earlier degradation of laminin coating, its beneficial effects on graft repopulation have been observed during the follow-up until week 8.”

Reviewer’s Comment 3

For unknown reasons, the supplemental figures are not available. Some of the supplemental figures should be presented in the main text, like the inflammation data, MMP data, since they are actually important.

Our Response:

We are very sorry that the supplemental figures have not been provided for the initial review process. Following the reviewer’s suggestion, our revised manuscript presents the figures on inflammation and MMP activity in the main text, so that the supplement contains only 2 figures.

Reviewer 2 Report

Authors present an study of the influence of Laminin for incresing the recellularization and reducing the hiperplasia. Results are concise and are well presented. Althougth more work need to be done in this regard, results shows a promising approach to improve the biocompatibility of tissue-engineered vascular implants.

Author Response

Response to Reviewer 2

Reviewer’s Comment 1

Authors present an study of the influence of Laminin for incresing the recellularization and reducing the hiperplasia. Results are concise and are well presented. Althougth more work need to be done in this regard, results shows a promising approach to improve the biocompatibility of tissue-engineered vascular implants.

Our Response:

We thank the reviewer for appreciation of the quality and presentation of our data. According to his suggestion, the manuscript has been reviewed and corrected by a native speaker.

Reviewer 3 Report

Materials and Methods section: I’d recommend subdividing with subheadings.

Results section doesn’t seem to contain sufficient information. Results shown in Figures should be reiterated in details for better understanding.

Section 3.2 (page 4, line 156), it is mentioned that the intensity was reduced; can authors provide how much reduction was observed based on fluorescence intensity?

Figure 1. Scale bar in A shows 500 but in the legend it shows 100um.

Figure 1. it looks like DAPI stained nuclei are visible only in 1D. If it’s the case, it should be mentioned.

Section 3.3. “obvious cellular colonization” it is not very obvious, perhaps arrows can be included in figures to indicate cellular colonizations. Was the decellularized aorta (shown in Figure 2D) implanted in vivo as well?

Figure 2. Representative REM -> SEM

Page 6, line 185 (Figure 4) may be Figure 5.

Page 6, line 202, it says figure 5 but it seems like it’s Figure 6.

Section 3.4. It is not very clear how the number of cells repopulating the media was measured.

Overall, please check the consistency throughout the manuscript, such as spacing between units and numbers and check for spelling.

Author Response

Response to Reviewer 3

Reviewer’s Comment 1

Materials and Methods section: I’d recommend subdividing with subheadings.

Our Response:

We completely agree with the reviewer that the M&M section should be structured this way – which we provide in the revision.

Reviewer’s Comment 2

Results section doesn’t seem to contain sufficient information. Results shown in Figures should be reiterated in details for better understanding.

Our Response:

Following the reviewer’s advice, we have added more information from the figures to the text in the results section, such as:

Section 3.3: “After 2 weeks, only extracellular matrix was detected on the luminal surface of SEM samples from explanted decellularized grafts, while autologous cellular colonization was not observed.” Section 3.3: “After 8 weeks in vivo, a continuous de novo cellular repopulation on the luminal side of decellularized conduits was observed, which was significantly higher in the laminin group (Percentage of re-endothelialized luminal surface: 98.4% ± 0.6% vs. 91.3% ± 3.1% in the control group, p=0.0048).” Section 3.3: “For analysis of adverse hyperplastic neointima formation, the intima-to-media ratio was calculated in explanted decellularized aortic grafts. After 8 weeks in vivo, the overall intima-to-media ratio in the laminin group was significantly lower than in the uncoated controls (0.9 ± 0.1 vs. 1.5 ± 0.2, p=0.0149) (Figure 4). In all subregions of the graft except in B1, the intima-to-media ratio in the laminin group was decreased as compared to controls.” Section 3.3: “Immunofluorescence staining of decellularized aortic grafts 8 weeks after implantation revealed most of the cells in the hyperplastic intima areas of both groups to contain aSMA. The luminal neoendothelial layer stained positive for vWF [Figure 5A,B]. By Movat’s pentachrome staining, in hyperplastic intima areas, glycosaminoglycan-rich extracellular substance was detected around cells with a fibroblastoid phenotype. In both groups, inflammatory cell markers (CD3 for lymphocytes and CD68 for macrophages) proved negative at 8 weeks [Figure 5C,D].” Section 3.3: “After 8 weeks in vivo, von Kossa staining revealed small areas of microcalcification in the neointima and in the tunica media and macrocalcification in the tunica media of the decellularized implants [Supplemental Figure S2]. Remodeled regions of the tunica media with high content of aSMa-positive autologous cells did not exhibit hydroxyapatite deposition. The extent of calcification, as assessed by the previously established von Kossa score, was not significantly different between the laminin and control groups, neither in the neointima (0.3 ± 0.1 in the laminin group vs. 0.5 ± 0.01 in the control group, p=0.0661), nor in the media (1.7 ± 0.2 vs. 1.1 ± 0.2, p=0.0779) (Figure 6).” Section 3.4: “The number of autologous cells repopulating the decellularized media was counted in all graft regions, and did not differ between the groups (Laminin: 127.6 ± 69.13 vs. Control: 242.2 ± 71.94 cells per cross-section, p=0.2577). In the laminin group, remodeling of the media by autologous cells migration and extracellular matrix production was mostly observed in the A1 and B2 regions around the graft anastomoses, while in the control group, it was found more frequently only in the A1 region. The restructured media was full of fibroblast-shaped cells.” Section 3.4: “In situ zymography after 8 weeks showed MMP activity in the adventitia and predominantly in the neointima of decellualrized aortic grafts. Areas of increased cell density, such as in hyperplastic neointima, exhibited a marked MMP activity, but there was no statistically significant difference between the two groups (p=0.0999) [Figure 7].”

Reviewer’s Comment 3

Section 3.2 (page 4, line 156), it is mentioned that the intensity was reduced; can authors provide how much reduction was observed based on fluorescence intensity?

Our Response:

Unfortunately, the fluorescence intensity has not been calculated. Although it may be interesting to get a fluorescence loss curve over time, we believe that intensity values would not add important information, not at last as the kinetics of fluorophore in vivo degradation and potential uncoupling are unknown. Therefore, we have rewritten the results and interpretation to present our findings of laminin persistency more carefully.

Please see results section 3.2 and the second paragraph of the discussion: “For interpretation of the study results, the durability of laminin coating is important. In earlier reports, there was no information on the persistency of laminin coating in vivo, whereas in our study, laminin-bound fluorescence was found to persist on decellularized grafts for at least 8 weeks. The green-fluorescent Alexa 488 coupled to the laminin proteins was initially observed along the surfaces of the grafts, however, after implantation in the blood circulation of rats, laminin has spread through the whole graft wall, which might be influenced by blood pressure. Besides laminin movement through the tissue, separation and movement of the fluorophores only may be considered as an explanation for the observed distribution of the fluorescence signal over time. In this scenario, laminin might have been degraded earlier, while the fluorophore might persist in the graft wall. On the other side, laminin might persist to an even larger extent than it was assumed from the detected fluorescence signal, since the fluorophore itself might have been degraded in vivo by enzymatic activity. Taken together, the kinetics of the in vivo degradation of fluorophores themselves as well as their uncoupling from bioactive proteins need to be further elucidated. Laminin antibody staining is not supportive in this context. We had previously conducted laminin antibody staining of decellularized and coated grafts, but all the grafts had stained positive, confirming data from other groups [6]. A reason for this issue is that most laminin isoforms are immunologically related to laminin 111, since they contain either the β1 or the δ1 chains, or both. It explains why antisera raised against laminin 111 purified from the EHS sarcoma stain all basement membranes, even if the laminin α1 chain is absent. However, even in case of potentially earlier degradation of laminin coating, its beneficial effects on graft repopulation have been observed during the follow-up until week 8.”

Reviewer’s Comment 4

Figure 1. Scale bar in A shows 500 but in the legend it shows 100um.

Figure 1. it looks like DAPI stained nuclei are visible only in 1D. If it’s the case, it should be mentioned.

Our Response:

Thank you very much for this correction. The scale bar and DAPI information have been adapted.

Reviewer’s Comment 5

Section 3.3. “obvious cellular colonization” it is not very obvious, perhaps arrows can be included in figures to indicate cellular colonizations. Was the decellularized aorta (shown in Figure 2D) implanted in vivo as well?

Our Response:

With the presented pictures, we wanted to show that “neither in the laminin group nor in the control group, obvious cellular colonization was observed”. In the revised version, we have reworded this sentence in section 3.3 (“After 2 weeks, only extracellular matrix was detected on the luminal surface of SEM samples from explanted decellularized grafts, while autologous cellular colonization was not observed.”), changed the subfigure showing a native aorta in Figure 2, and inserted arrows to improve the clarity of the data.

Reviewer’s Comment 6

Figure 2. Representative REM -> SEM

Our Response:

Thank you very much for this correction.

Reviewer’s Comment 7

Page 6, line 185 (Figure 4) may be Figure 5.

Our Response:

Thank you very much for this correction, the figure numbers were mixed.

Reviewer’s Comment 8

Page 6, line 202, it says figure 5 but it seems like it’s Figure 6.

Our Response:

Thank you very much for this correction, the figure numbers were mixed.

Reviewer’s Comment 9

Section 3.4. It is not very clear how the number of cells repopulating the media was measured.

Our Response:

In order to clarify our measuring method, we have extended the methods section (2.6 Histology) as follows: “In brief, each graft was divided into four regions as described above and three sections from each region were divided into eight segments by radial lines (angle: 45°), originating from the center of the aortic lumen. … Similarly, in each of the 8 graft wall segments, the repopulation of the media of the grafts was evaluated as follows: In each cross-section, the total number of cells invading the medial part of the decellularized implants was counted.”

The results section 3.4 has been expanded as follows: “The number of cells repopulating the media was counted in all graft regions, and did not differ between the groups (127.6 ± 69.13 vs. 242.2 ± 71.94 cells per cross-section, p=0.2577).”

Reviewer’s Comment 10

Overall, please check the consistency throughout the manuscript, such as spacing between units and numbers and check for spelling.

Our Response:

Thank you very much for bringing this to our attention. Spacing between units and numbers as well as spelling have been checked throughout the manuscript by a native speaker.

Round 2

Reviewer 1 Report

All the issues have been addressed